# Cross Pseudo Supervision for Semi-supervised Medical Image Segmentation

Peng Liu and Guoyan Zheng

Shanghai Jiao Tong University, Shanghai, China

**Abstract.** Multi-organ segmentation is an important step in many medical image applications. Due to time-consuming and challenging for physicians to annotate multi-organ segmentation dataset, the existing dataset for multi-organ segmentation usually have small number of samples. A series of works have been proposed to make use of these limited annotated data for improving the performance of multi-organ segmentation. In this paper, In this paper, we present a novel context-aware cross pseudo supervision algorithm for semi-supervised medical image segmentation. Our method first use two networks with different initialization strategies, then we fed two overlapped patches to the network, last we use the outputs of the overlapped regions of one network to get the pseudo label to supervised another network. Experimental Results show that our proposed method perform well in multi-organ segmentation.

**Keywords:** Semi-supervised Learning · Multi-organ segmentation · Cross Pseudo Supervision.

## 1 Introduction

Multi-organ segmentation is an essential step in many clinical practices including diagnostic interventions, surgical planning and treatment delivery[5]. Recently, with the arise of deep learning in computer vision domain, researchers in medical image analysis also use deep learning to do their tasks and achieve great improvement. For deep learning in multi organ segmentation, most of the works are based on deep convolutional neural network(DCNN)[8,13,11,12,1,2]. These methods achieve great success. However, this success primarily relies on the large-scale labeled dataset. In comparison, annotating large scale medical image dataset is time-consuming and usually need highly-trained physicians. In order to reduce the workload of annotating a large scale dataset, some physicians may choose to annotate few images. Therefore, how to make use of large scale unlabeled dataset to train a better multi-organ segmentation model has become a practical problem. This makes semi-supervised segmentation an important problem to learn segmentation models by using the labeled data as well as the additional unlabeled data.

Consistency regularization is widely studied in semi-supervised semantic segmentation. It enforces the consistency of the predictions with various perturbations.We adopt a simple consistency regularization approach with network perturbation, called cross pseudo supervision[3]. The proposed approach feeds the

labeled and unlabeled images into two segmentation networks that share the same structure and are initialized differently. The outputs of the two networks on the labeled data are supervised separately by the corresponding ground-truth segmentation map.

## 2   Method

The proposed approach consists of two parallel segmentation networks. Both two networks are 3D U-Net.

### 2.1   Preprocessing

Our patch size is [160, 160, 96]. We cut the intensity into -68 200.We use Z-score Normalization for the data.

### 2.2   Proposed Method

The overall architecture of our method is shown in Figure 1. We use two parallel segmentation networks based on 3D U-Net.

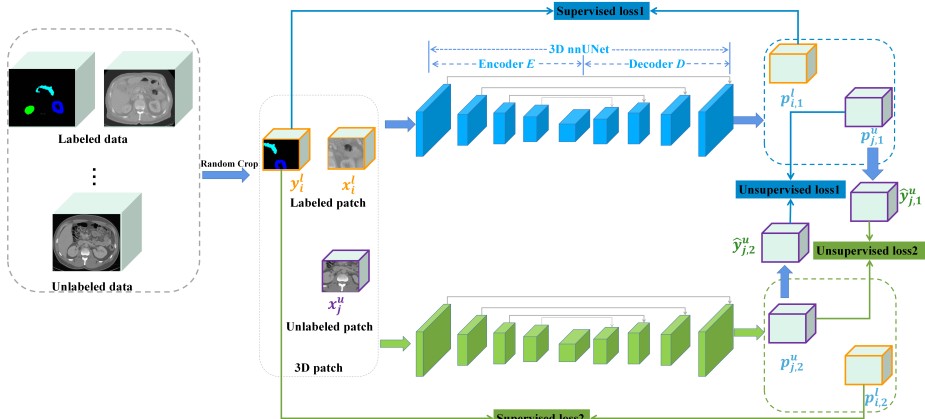

**Fig. 1.** Network architecture

We use cross pseudo label to leverage the unlabeled data.

Network architecture details

Loss function: we use the summation between Dice loss and cross entropy loss because compound loss functions have been proved to be robust in various medical image segmentation tasks [9].

Strategies to improve inference speed and reduce resource consumption (Based on the winning solutions in FLARE 2021, we recommend using ONNX or TensorRT to speed up inference process)

### 2.3   Post-processing

## 3   Experiments

### 3.1   Dataset and evaluation measures

The FLARE2022 dataset is curated from more than 20 medical groups under the license permission, including MSD [14], KiTS [6,7], AbdomenCT-1K [10], and TCIA [4]. The training set includes 50 labelled CT scans with pancreas disease and 2000 unlabelled CT scans with liver, kidney, spleen, or pancreas diseases. The validation set includes 50 CT scans with liver, kidney, spleen, or pancreas diseases. The testing set includes 200 CT scans where 100 cases has liver, kidney, spleen, or pancreas diseases and the other 100 cases has uterine corpus endometrial, urothelial bladder, stomach, sarcomas, or ovarian diseases. All the CT scans only have image information and the center information is not available.

The evaluation measures consist of two accuracy measures: Dice Similarity Coefficient (DSC) and Normalized Surface Dice (NSD), and three running efficiency measures: running time, area under GPU memory-time curve, and area under CPU utilization-time curve. All measures will be used to compute the ranking. Moreover, the GPU memory consumption has a 2 GB tolerance.

### 3.2   Implementation details

**Environment settings** The development environments and requirements are presented in Table 1.

**Table 1.** Development environments and requirements.

| | |
|---|---|
| Windows/Ubuntu version | Ubuntu 18.04.5 LTS |
| CPU | Intel(R) Xeon(R) Platinum 8168 CPU@2.70GHz |
| RAM 1.5T | |
| GPU (number and type) | NVIDIA Tesla V100 |
| CUDA version | 11.0 |
| Programming language | Python 3.7 |
| Deep learning framework | Pytorch (Torch 1.60) |
| Specific dependencies | |
| (Optional) Link to code | |

**Training protocols** Please describe at least the following aspects:

Data augmentation (Based on the winning solutions in FLARE 2021, we recommend using extensive data augmentation) patch sampling strategy, optimal model selection criteria

**Table 2.** Training protocols.

| | |
|---|---|
| Network initialization | "he" normal initialization |
| Batch size | 2 |
| Patch size | 96×160×160 |
| Total epochs | 1000 |
| Optimizer | SGD with nesterov momentum ($\mu = 0.99$) |
| Initial learning rate (lr) | 0.01 |
| Lr decay schedule | halved by 200 epochs |
| Training time | 96 hours |
| Number of model parameters | |
| Number of flops | |
| $CO_2$eq | |

## 4    Results and discussion

Note: Please describe at least the following aspects:
The effect of using unlabelled cases;
What kind of cases the proposed method works well?
What are the possible reasons for the failed cases or organs?
Segmentation efficiency analysis

### 4.1    Quantitative results on validation set

Currently, you can report the Dice score on validation set
    Please do ablation study to analysis the effect of unlabelled data.

### 4.2    Qualitative results on validation set

This part is optional during validation phase since you do not have validation ground truth.

### 4.3    Segmentation efficiency results

### 4.4    Limitation and future work

## 5    Conclusion

The main finding and results

**Acknowledgements** The authors of this paper declare that the segmentation method they implemented for participation in the FLARE 2022 challenge has not used any pre-trained models nor additional datasets other than those provided by the organizers. The proposed solution is fully automatic without any manual intervention.

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
