# OpenReview forum: "Cross Pseudo Supervision for Semi-supervised Medical Image Segmentation"
_MICCAI.org/2022/Challenge/FLARE_

### Official Review · Reviewer_3oui · 2022-09-13
**The idea looks nice but the paper is incomplete.**

**Rating:** 2
**Confidence:** 4

**Review:**

Pros: The figure of the network architecture looks nice.

Cons:
1. It seems that there are some incomplete sections, and the details of the pipeline remains unclear.

---

### Official Review · Reviewer_j3it · 2022-09-13
**This paper is left unfinished**

**Rating:** 2
**Confidence:** 5

**Review:**

Authors apply Cross Pseudo Supervision (CPS) into a pipeline that consists of two parallel 3D U-Nets.

This paper is left unfinished. Only some information are given in the preprocessing and proposed method sections.

---

### Official Review · Reviewer_F8Ym · 2022-09-19
**A method use cross pseudo surpervision for semi-supervised segmentation**

**Rating:** 2
**Confidence:** 4

**Review:**

The authors present a novel context-aware cross pseudo supervision algorithm for semi-supervised medical image segmentation. The method first use two networks with different initialization strategies, then fed two overlapped patches to the network. At last it use the outputs of the overlapped regions of one network to get the pseudo label to supervised another network.

There are some suggestions:
1. The article should reflect the specific indicators on the validation set(DSC, NSD,...) in the abstract.
2. The article has too few description on methods and preprocess course.
3. The article should give more explanation to Figure 1.
4. The article lacks important part: results and discussions (4), quantitative results (4.1), qualitative results (4.2), segmentation efficiency results (4.3), limitations and future work (4.4) and conclusions (5).
5. The article lacks necessary parts. Please confirm whether the wrong file has been uploaded?

---

### Meta-Review · Program_Chairs · 2022-09-28

**Recommendation:** Major Revision
**Confidence:** 5

**Metareview:**

Reviewers raise many concerns and suggestions. Please address all comments in the revised manuscript.